# Feudal Graph Reinforcement Learning

**Tommaso Marzi**                                          *tommaso.marzi@usi.ch*
*Università della Svizzera italiana, IDSIA*

**Arshjot Khehra**                                          *arshjot.khehra@usi.ch*
*Università della Svizzera italiana*

**Andrea Cini**                                          *andrea.cini@usi.ch*
*Università della Svizzera italiana, IDSIA*

**Cesare Alippi**                                          *cesare.alippi@usi.ch*
*Università della Svizzera italiana, IDSIA*
*Politecnico di Milano*

**Reviewed on OpenReview:** *https://openreview.net/forum?id=wFcyJTik90*

## Abstract

Graph-based representations and message-passing modular policies constitute prominent approaches to tackling composable control problems in reinforcement learning (RL). However, as shown by recent graph deep learning literature, such local message-passing operators can create information bottlenecks and hinder global coordination. The issue becomes more serious in tasks requiring high-level planning. In this work, we propose a novel methodology, named *Feudal Graph Reinforcement Learning* (FGRL), that addresses such challenges by relying on hierarchical RL and a pyramidal message-passing architecture. In particular, FGRL defines a hierarchy of policies where high-level commands are propagated from the top of the hierarchy down through a layered graph structure. The bottom layers mimic the morphology of the physical system, while the upper layers correspond to higher-order sub-modules. The resulting agents are then characterized by a committee of policies where actions at a certain level set goals for the level below, thus implementing a hierarchical decision-making structure that can naturally implement task decomposition. We evaluate the proposed framework on a graph clustering problem and MuJoCo locomotion tasks; simulation results show that FGRL compares favorably against relevant baselines. Furthermore, an in-depth analysis of the command propagation mechanism provides evidence that the introduced message-passing scheme favors learning hierarchical decision-making policies.

## 1 Introduction

Although reinforcement learning (RL) methods paired with deep learning models have recently led to outstanding results, e.g., see (Silver et al., 2017; Degrave et al., 2022; Wurman et al., 2022; Ouyang et al., 2022), achievements have come with a severe cost in terms of sample complexity. A possible way out foresees embedding inductive biases into the learning system, for instance, by leveraging the relational compositionality of the tasks and physical objects involved (Battaglia et al., 2018; Hamrick et al., 2018; Zambaldi et al., 2018; Funk et al., 2022a). Within this context, physical systems can be often decomposed into a collection of discrete entities interconnected by binary relationships. In such cases, graphs emerge as a suitable representation to capture the system's underlying structure. When processed by message-passing graph neural networks (GNNs) (Gilmer et al., 2017; Bacciu et al., 2020; Bronstein et al., 2021), these graph representations enable the reuse of experience and the transfer of models across agents (Wang et al., 2018): node-level modules (policies) can easily be applied to graphs (systems) with different topologies (structures). However, while conceptually appealing, the use of modular message-passing policies also bears some concerns in terms

of the constraints that such representations impose on the learning system. As an example, consider a robotic agent: a natural relational representation can be obtained by considering a graph capturing its morphology, with links represented as edges, and different types of joints (or limbs) as nodes. In this framework, policies can be learned at the level of the single decision-making unit (actuator, in the robotic example), in a distributed fashion (Wang et al., 2018; Huang et al., 2020). Nevertheless, recent evidence suggests that existing approaches are not successful in learning composable policies able to fully exploit agents' morphology (Kurin et al., 2020). Notably, in such modular architectures, the same replicated processing module must act as a low-level controller (e.g., by applying a torque to a joint of a robotic leg), but, at the same time, attend to complex, temporally extended tasks at the global level, such as running or reaching a far-away goal.

In light of this, we propose to tackle the problem of high-level coordination in modular architecture introducing a novel hierarchical approach to designing graph-based message-passing policies. Our approach is inspired by the guiding principles of hierarchical reinforcement learning (HRL) (Barto & Mahadevan, 2003) and, in particular, feudal reinforcement learning (FRL) (Dayan & Hinton, 1992): we argue that GNNs are natural candidates to implement HRL algorithms, as their properties implicitly account for HRL desiderata such as transferability and task decomposition. Indeed, one of the core ideas of HRL is structuring the learning systems to facilitate reasoning at different levels of spatiotemporal abstraction. The options framework (Sutton et al., 1999), as an example, pursues temporal abstraction by expanding the set of available actions with temporally extended behavioral routines, implemented as policies named options. Conversely, FRL implements spatiotemporal abstractions by relying on a hierarchy of controllers where policies at the top levels set the goals for those at lower levels. In the FRL framework, each decision-making unit is seen as a manager that controls its sub-managers and, in turn, is controlled by its super-manager. In its original formulation, FRL is limited to tabular RL settings and relies on domain knowledge for the definition of subtasks and intermediate goals.

Although extensions to the deep RL settings exist (Vezhnevets et al., 2017), the idea of learning a hierarchy of communicating policies has not been fully exploited yet. In this paper, we rely on these ideas to propose a novel hierarchical graph-based methodology, named *Feudal Graph Reinforcement Learning* (FGRL), to build hierarchical committees of composable control policies. In FGRL, policies are organized within a feudal, i.e., pyramidal, structure, and each layer corresponds to a graph. Message-passing GNNs provide the proper neuro-computational framework to implement the architecture. More in detail, the hierarchy of controllers is represented as a multilayered graph where nodes at the bottom (workers) can be seen as actuator-level controllers, while upper-level nodes (sub-managers and manager) can focus on high-level planning, possibly by exploiting higher-order relationships. In the robotic example, workers would correspond to the agent's joints and the corresponding actuators, i.e., to those entities directly applying a control action to the system. Conversely (sub-)managers would correspond to modules responsible for coordinating subordinate nodes by sending appropriate (possibly high-level) commands. Depending on their role in the hierarchy, such commands might specify, for instance, a target position for a specific joint or control the orientation of a body part. In FGRL, each discrete decision-making unit is responsible for the control of the system at a certain level. In particular, each node participates in learning a policy by exchanging messages with controllers of the same levels and setting local goals for the level below. While exchanged messages enable coordination, the hierarchical pyramidal structure constrains the information flow, implementing information hiding. As a result, the system is biased towards learning a hierarchical decomposition of the problem into sub-tasks.

To summarize, our main novel contributions are as follows.

1. We introduce the FGRL paradigm, a new methodological deep learning framework for graph-based HRL in composable environments (Sec. 4).

2. We evaluate a possible implementation of the proposed method on a graph clustering problem and on composable continuous control tasks from the MuJoCo locomotion benchmarks (Todorov et al., 2012), where the proposed approach obtains competitive performance w.r.t. relevant baselines for composable control (Sec. 5.2 and 5.3).

3. We provide empirical evidence that supports the adoption of hierarchical message-passing schemes and graph-based representations to implement hierarchical control policies (Sec. 5.2 and 5.3).

Our work paves the way for a novel take on hierarchical and graph-based reinforcement learning, marking a significant step toward designing deep RL architectures incorporating biases aligned with the structure of HRL agents.

The paper is structured as follows. Sec. 2 discusses related works. The FGRL framework is introduced in Sec. 4 and empirically evaluated in Sec. 5. Sec. 6 reports final considerations and discusses directions for future works.

## 2 Related Works

Several RL methods rely on relational representations. Zambaldi et al. (2018) embed relational inductive biases into a model-free deep RL architecture by exploiting the attention mechanism. Sanchez-Gonzalez et al. (2018) use GNNs to predict the dynamics of simulated physical systems and show applications of such models in the context of model-based RL. Other works adopt GNNs in place of standard, fully-connected, feed-forward networks to learn policies and value functions for specific structured tasks, such as physical construction (Hamrick et al., 2018; Bapst et al., 2019). Moreover, GNNs have been exploited also in the context of multi-agent systems (Jiang et al., 2020) and robotics (Funk et al., 2022a;b). Ha & Tang (2022) provide an overview of deep learning systems based on the idea of collective intelligence, i.e., systems where the desired behavior emerges from the interactions of many simple (often identical) units.

More related to our approach, NerveNet (Wang et al., 2018) relies on message passing to propagate information across nodes and learn an actuator-level policy. Similarly to NerveNet, the *shared modular policies* (SMP) method (Huang et al., 2020) learns a global policy that is shared across the limbs of a target agent and controls simultaneously different morphologies. The agents' structure is encoded by a tree where an arbitrary limb acts as a root node. Information is propagated through the tree in two stages, from root to leaves and then backward. Kurin et al. (2020), however, show that constraining the exchange of information to the structure of the system being controlled can hinder performance. This issue is a well-known problem in graph machine learning: the best structure to perform message passing does not necessarily correspond to the input topology (Topping et al., 2022; Rusch et al., 2023). Graph pooling (Ying et al., 2018; Bianchi et al., 2020; Grattarola et al., 2022; Bianchi & Lachi, 2024) tackles this problem by clustering nodes and rewiring the graph to learn hierarchical representations. Our work can be seen as introducing a similar idea in graph-based RL. Moreover, in FGRL, the hierarchical structure corresponds to the structure of the decision-making process. A different research line on composable RL leverages Transformers (Vaswani, 2017) to learn universal policies designed for handling multiple morphologies simultaneously (Trabucco et al., 2022; Gupta et al., 2022; Xiong et al., 2023), possibly encoding geometric symmetries in the model (Chen et al., 2023). In this context, Xiong et al. (2024) propose an approach to improve the efficiency of existing multi-task RL methods by generating policies using a morphology-conditioned hypernetwork (Ha et al., 2017).

FRL (Dayan & Hinton, 1992) has been extended to the deep RL setting with the introduction of *FeUdal Networks* (FUN) (Vezhnevets et al., 2017) and explored in the context of multi-agent systems (Ahilan & Dayan, 2019). However, none of the previous works match FRL with a hierarchical graph-based architecture to learn modular policies, as we do here instead.

## 3 Preliminaries

**Markov decision processes** A *Markov decision process* (MDP) (Sutton & Barto, 2018) is a tuple $\langle \mathcal{S}, \mathcal{A}, \mathcal{P}, \mathcal{R} \rangle$ where $\mathcal{S} \subseteq \mathbb{R}^{n_s}$ is a state space, $\mathcal{A} \subseteq \mathbb{R}^{n_a}$ is an action space, $\mathcal{P} : \mathcal{S} \times \mathcal{A} \to \mathcal{S}$ is a Markovian transition function and $\mathcal{R} : \mathcal{S} \times \mathcal{A} \to \mathbb{R}$ is a payoff (reward) function. We focus on the *episodic* RL setting where the agent acts in the environment for a fixed number of time steps or until it receives a termination signal from the environment. The objective is to learn a parameterized stochastic policy $\pi_\theta$ that maximizes the total expected reward received in an episode. We focus on environments where the state representation can be broken down into sub-parts, each part mapped to a node of a graph structure.

**Graphs and message-passing neural networks** A graph is a tuple $\mathcal{G} = \langle \mathcal{V}, \mathcal{E} \rangle$, where $\mathcal{V}$ and $\mathcal{E}$ denote the set of vertices (nodes) and links (edges), respectively. In attributed graphs, each node $i$ is equipped

with an attribute (or feature) vector $\boldsymbol{x}_i \in \mathbb{R}^{d_x}$. Similarly, edges can be associated with attribute vectors $\boldsymbol{e}_{ij} \in \mathbb{R}^{d_e}$ where $(i, j)$ indicates the edge connecting the $i$-th and the $j$-th nodes. Message-passing neural networks (MPNNs) (Gilmer et al., 2017) encompass a large variety of GNN architectures under the same general framework. In particular, in MPNNs, representations associated with each node are updated at each round by aggregating messages from its neighbors. More precisely, representation $\boldsymbol{x}_i^l$ of node $i$ at round $l$ with neighbors $\mathcal{N}(i)$ is updated as:

$$\boldsymbol{x}_i^{l+1} = \zeta^l \left( \boldsymbol{x}_i^l, \operatorname*{AGGR}_{j \in \mathcal{N}(i)} \left\{ \phi^l(\boldsymbol{x}_i^l, \boldsymbol{x}_j^l, \boldsymbol{e}_{ij}) \right\} \right), \tag{1}$$

where AGGR is a permutation-invariant aggregation function, while $\zeta^l$ and $\phi^l$ are, respectively, differentiable update and message functions, e.g., implemented by multilayer perceptrons (MLPs).

## 4 Feudal Graph Reinforcement Learning

A physical system can often be described as a set of entities and relationships among these entities. As a case study, we consider structured agents (simplified robots), as those typically used in continuous control RL benchmarks (Tassa et al., 2018). Such agents are modeled as made of several joints and limbs that correspond to actuators of a system to control. The following section provides a method for extracting and exploiting hierarchical graph-based representations in RL.

### 4.1 Graph-based Agent Representation

A structured agent with $K$ limbs can be represented as an undirected graph $\mathcal{G}_1$. The subscript denotes the level in the hierarchy and will be contextualized in the next subsection. In this setup, each $i$-th limb with $i \in \{1, \dots, K\}$ is mapped to a node whose feature vector $\boldsymbol{s}_i^t \in \mathcal{S}$ contains information regarding its state at time $t$ (e.g., the position, orientation, and velocity of the node); an edge between two nodes indicates that the corresponding limbs are connected. Each limb can be paired with an actuator and outputs are associated with control actions $\boldsymbol{a}_i^t \in \mathcal{A}$. Limbs with no associated action act as auxiliary hub nodes and simply account for the morphology of the agent; in practice, representations of such nodes

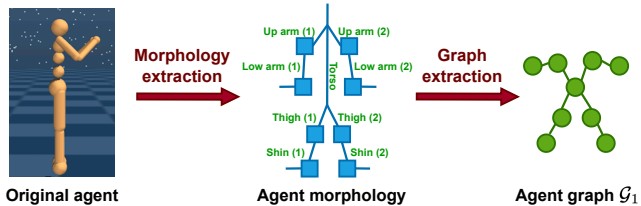

Figure 1: Constructing the agent graph $\mathcal{G}_1$ for 'Humanoid' environment. Blue squares in the agent's morphology represent the joints of the agent and are not mapped to nodes, differently from the green labels which, instead, refer to the limbs and constitute the nodes of $\mathcal{G}_1$.

are simply not mapped to actions and are discarded in the final processing steps. Fig. 1 provides an example of the graph extraction process for the 'Humanoid' environment from MuJoCo, in which the *torso* node acts as a simple hub for message passing.

### 4.2 Building the Feudal Hierarchy

The core idea of FGRL consists of exploiting a multi-level hierarchical graph structure $\mathcal{G}^*$ to model and control a target system by leveraging a pyramidal decision-making architecture. In this framework, each $i$-th node in the hierarchy reads from the representation of subordinate (child) nodes $\mathcal{C}(i)$ and assigns them goals; in turn, it is subject to goals imposed by its supervisor (parent) nodes $\mathcal{P}(i)$ through the same goal-assignment mechanism. Each node has no access to state representations associated with the levels above, as the structure is layered. We identify three types of nodes:

1. **Manager**: It is a single node at the highest level of the hierarchy, i.e., it has no supervisor. It receives the reward directly from the environment and is responsible for coordinating the entire hierarchy.

2. **Sub-managers**: These nodes constitute the intermediate levels of the hierarchy and, for each level $l_h$, they can interact among each other through message passing on the associated graph $\mathcal{G}_{l_h}$. Each sub-manager receives an intrinsic reward based on the goals set by its direct supervisors.

3. **Workers**: These nodes are at the lowest level of the hierarchy. Workers correspond to nodes in the previously introduced base graph $\mathcal{G}_1$ and are in charge of directly interacting with the environment. At each time step $t$, the $i$-th worker has access to state vector $\boldsymbol{s}_i^t$ and receives an intrinsic reward that is a function of its assigned goals.

This feudal setup allows for decomposing the original task into simpler sub-tasks by providing goals at different scales. In particular, increasing the hierarchy depth $L_h$ increases the resolution at which goals are assigned: upper levels can focus on high-level planning, while lower levels can set more immediate goals. The resulting hierarchical committee facilitates temporal abstraction, which can be further promoted by letting each level of the hierarchy act at different temporal scales (Vezhnevets et al., 2017). In practice, we can keep the commands set by the higher levels fixed for a certain (tunable) number of steps. Depending on the setup, sub-managers and workers can rely on intrinsic rewards alone (reward hiding principle; Dayan & Hinton (1992)) or maximize both intrinsic and extrinsic reward at the same time (Vezhnevets et al., 2017).

The hierarchical graph $\mathcal{G}^*$ can be built by clustering nodes of the base graph $\mathcal{G}_1$ (with nodes possibly belonging to more than one cluster) and assigning each group to a sub-manager. The same clustering process can be repeated to obtain a single manager at the top of the hierarchy; see Fig. 2 (left) and Fig. 7 for reference. We remark that our study focuses on composable control problems in structured environments, i.e., where agents can be decomposed into interconnected functional units. In this setting, the base graph and, possibly, the hierarchy can be easily inferred. Indeed, clustering can be performed in several ways, e.g., by grouping nodes according to physical proximity or functionality (e.g., by aggregating actuators belonging to a certain subcomponent). Nevertheless, when heuristics do not allow for extracting the hierarchy, graph pooling (Grattarola et al., 2022; Bianchi & Lachi, 2024) provides a set of robust and well-understood operators for constructing the hierarchical graph. We indicate with $\mathcal{G}_{l_h}$ the pooled graph representation at the $l_h$-th level of the hierarchy and remark that the state of each node is hidden from entities at lower levels. The number of hierarchy levels is a hyperparameter that depends on the problem at hand.

### 4.3 Learning Architecture

Given the introduced hierarchical setup, this subsection illustrates the learning architecture and the procedure to generate actions starting from the environment state. In particular, we generate the initial representations of nodes in $\mathcal{G}^*$ starting from raw observations, in a bottom-up fashion. Subsequently, to take full advantage of both the feudal structure and the agent's morphology, information is propagated across nodes at the same level as well as through the hierarchical structure. Finally, (sub-)managers set goals top-down through $\mathcal{G}^*$, while workers act according to the received commands. We now break this process down into a step-by-step procedure, a visual representation of which is reported in Fig. 2.

**State representation** The environment state is partitioned and mapped to a set $\{\boldsymbol{s}_i\}_{i=1}^K$ of $K$ local node states, each corresponding to an actuator. We omit the temporal index $t$ as there is no ambiguity. Additional (positional) node features $\{\boldsymbol{f}_i\}_{i=1}^K$ can be included in the representation, thus generating observation vectors $\{\boldsymbol{x}_i\}_{i=1}^K$, where, for each $i$-th limb, $\boldsymbol{x}_i$ is obtained by concatenating $\boldsymbol{s}_i$ and $\boldsymbol{f}_i$. Starting from the observation vectors, the initial representation $\boldsymbol{h}_i^0$ of each $i$-th node in $\mathcal{G}^*$ is obtained recursively as:

$$\boldsymbol{h}_i^0 = \begin{cases} \boldsymbol{W}_1 \boldsymbol{x}_i, & i \in \mathcal{G}_1 \\ \underset{j \in \mathcal{C}(i)}{\text{AGGR}} \left\{ \rho^{l_h}(\boldsymbol{h}_j^0) \right\}, & i \in \mathcal{G}_{l_h}, \ l_h \in \{2, \dots, L_h\} \end{cases} \tag{2}$$

where $\boldsymbol{W}_1 \in \mathbb{R}^{d_h \times d_x}$ is a learnable weight matrix, $\rho^{l_h}$ a (trainable) differentiable function and AGGR a generic aggregation function. The superscript of $\boldsymbol{h}_i^0$ indicates the stage of the information propagation. In practice, the initial state representation of each worker is obtained through a linear transformation of the corresponding observation vector, while, for (sub-)managers, representations are initialized bottom-up with a many-to-one mapping obtained by aggregating recursively representations at the lower levels of $\mathcal{G}^*$.

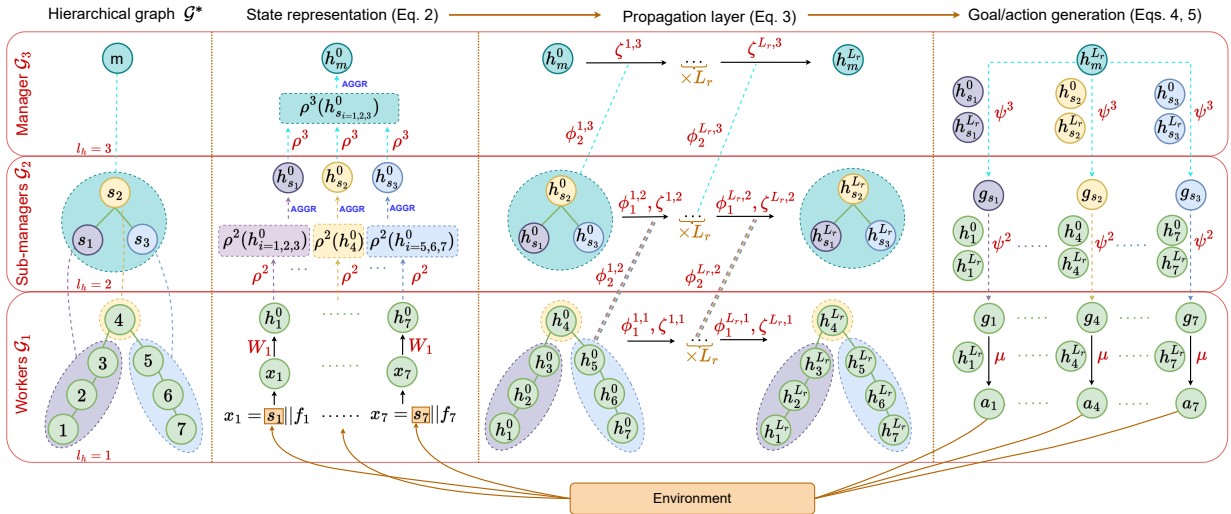

Figure 2: Learning architecture given the hierarchical graph $\mathcal{G}^*$ and the graphs $\mathcal{G}_{l_h}$ for the 'Walker' environment. Trainable functions are reported in red and hierarchical operations are represented with dashed lines: in $\mathcal{G}^*$, information flows bottom-up, while goals are assigned top-down.

**Propagation layer** Information across nodes is propagated through message passing. Nodes at the $l_h$-th level can read from representations of neighbors in $\mathcal{G}_{l_h}$ and subordinate nodes in $\mathcal{G}_{l_h-1}$, i.e., those corresponding to the lower level. We combine these 2 information flows in a single message-passing step. In particular, starting from the initial representations $\boldsymbol{h}_i^0$, each round $l_r \in \{1, \ldots, L_r\}$ of message passing updates the corresponding state representation $\boldsymbol{h}_i^{l_r}$ as:

$$\boldsymbol{h}_i^{l_r} = \zeta^{l_r, l_h}\left(\boldsymbol{h}_i^{l_r-1}, \underset{j \in \mathcal{N}(i)}{\mathrm{AGGR}}\left\{\phi_1^{l_r, l_h}(\boldsymbol{h}_i^{l_r-1}, \boldsymbol{h}_j^{l_r-1}, \boldsymbol{e}_{ij})\right\}, \underset{j \in \mathcal{C}(i)}{\mathrm{AGGR}}\left\{\phi_2^{l_r, l_h}(\boldsymbol{h}_i^{l_r-1}, \boldsymbol{h}_j^{l_r-1}, \boldsymbol{e}_{ij})\right\}\right), \quad (3)$$

where the message function $\phi_1^{l_r, l_h}$ regulates the exchange of information among nodes at the same hierarchy level, $\phi_2^{l_r, l_h}$ conveys, conversely, information from the subordinate nodes, and $\zeta^{l_r, l_h}$ updates the representation. We comment that workers (nodes at the lowest level in the hierarchy) only receive messages from neighbors in $\mathcal{G}_1$, while the top-level manager simply reads from its direct subordinates. We remark that, in general, for each round $l_r$ and hierarchy level $l_h$ we can have different message and update functions. However, multiple steps of message passing can lead to over-smoothing in certain graphs (Rusch et al., 2023). In such cases, the operator in Eq. 3 can be constrained to only receive messages from neighbors at the same level. Note that information nonetheless flows bottom-up at the initial encoding step (Eq. 2) and top-down through goals, as discussed below.

**Goal generation** State representations are used to generate goals (or commands) in a recursive top-down fashion through $\mathcal{G}^*$. In particular, each supervisor $i \in \mathcal{G}_{l_h}$, with $l_h \in \{2, \ldots, L_h\}$, sends a local goal $\boldsymbol{g}_{i \to j}$ to each subordinate node $j \in \mathcal{C}(i)$ as:

$$\boldsymbol{g}_{i \to j} = \begin{cases} \psi^{L_h}\left(\boldsymbol{h}_i^{L_r}, \boldsymbol{h}_j^{L_r}, \boldsymbol{h}_j^0\right), & i \in \mathcal{G}_{L_h} \\ \psi^{l_h}\left(\underset{k \in \mathcal{P}(i)}{\mathrm{AGGR}}\left\{\boldsymbol{g}_{k \to i}\right\}, \boldsymbol{h}_j^{L_r}, \boldsymbol{h}_j^0\right), & \text{otherwise} \end{cases} \quad (4)$$

where the superscript $L_r$ denotes the last round of message passing (see Eq. 3). We remark that the top-level manager has no supervisor: goals here are directly generated from its state representation, which encompasses the global state of the agent. All goal (or command) functions $\psi^{l_h}$ used by (sub-)managers can be implemented, for example, as MLPs, while worker nodes do not have an associated goal-generation mechanism, but, instead, have a dedicated action-generation network.

**Action generation**  Lowest-level modules map representations to raw actions, which are then used to interact with the environment. For each $i$-th node in $\mathcal{G}_1$, the corresponding action $\boldsymbol{a}_i$ is computed as a function of the aggregation of the goals set by the corresponding supervisors and, possibly, its state representation:

$$\boldsymbol{a}_i = \mu \left( \underset{j \in \mathcal{P}(i)}{\text{AGGR}} \{\boldsymbol{g}_{j \to i}\}, \boldsymbol{h}_i^{L_r} \right) \tag{5}$$

The action-generation function $\mu$ can be again implemented by, e.g., an MLP shared among workers. We remark that actions associated with nodes that do not correspond to any actuator are discarded.

**Rewards**  Each node of the hierarchy receives a different reward according to the associated goals and its role in the hierarchy. As already mentioned, the top-level manager coordinates the entire hierarchy and collects rewards directly from the environment. On the other hand, sub-managers and workers receive intrinsic rewards that can be used either as their sole reward signal or added to the external one. As an example, a dense reward signal for the $i$-th worker can be generated as a function of the received goals and the state transition:

$$r_i = f_R \left( \underset{j \in \mathcal{P}(i)}{\text{AGGR}} \{\boldsymbol{g}_{j \to i}\}, \boldsymbol{s}_i, \boldsymbol{s}_i' \right), \tag{6}$$

where $f_R$ is a score function and $\boldsymbol{s}_i'$ denotes the subsequent state. We remark that since goals are learned and are (at least partially) a function of $\boldsymbol{s}_i$, designing a proper score function $f_R$ is critical to avoid degenerate solutions. Nevertheless, the reward scheme outlined in Appendix D.2 and empirically validated in Appendix C.3 can be easily applied to many tasks without any prior knowledge. At each step, rewards of nodes belonging to the same level $l_h$ are combined and then aggregated over time to generate a cumulative reward (or return) $R_{l_h}$, which is subsequently used as a learning signal for that level.

**Scalability**  MPNNs are inductive, i.e., not restricted to process input graphs of a fixed size. As a result, the number of learning parameters mainly depends on the depth of the hierarchy, which is a hyperparameter that depends on the problem at hand. Nevertheless, we remark that, in graph deep learning literature, hierarchies with a small number of levels (e.g., 3 levels) have proved to be effective even when dealing with large graphs (Wu et al., 2022), so there is usually no need for using very deep hierarchies.

## 5  Experiments

This section introduces the experimental setup and compares FGRL against relevant baselines; furthermore, we provide an in-depth study of the goal-generation mechanism. Note that the main objective here is to compare FGRL against other composable architectures such as agents based on *flat*, i.e., not hierarchical, GNNs.

### 5.1  Experimental Setup

We validate our framework on two scenarios, namely a synthetic graph clustering problem inspired by Bianchi et al. (2020) and continuous control environments from the standard MuJoCo locomotion tasks (Todorov et al., 2012), where we follow Huang et al. (2020). More details on the graph clustering environment are reported in Appendix B.

**Compared approaches**  The proposed approach is compared against multiple variants thereof:

1. **Feudal graph neural network (FGNN)**: This is a complete implementation of the proposed framework.

2. **Feudal deep set (FDS)**: This is an implementation of our framework that does not exploit any message passing. FDS can be seen as a modular version of a hierarchical agent akin to Vezhnevets et al. (2017).

3. **Graph neural network (GNN)**: Similarly to NerveNet (Wang et al., 2018), this model performs message passing on the agent graph $\mathcal{G}_1$ (see Eq. 1), but does not exploit the hierarchical structure. An MLP then maps node-level representations to actions.

4. **Deep set (DS)**: This baseline models the agent as a set of entities each corresponding to a limb, but does not rely on any structure to propagate representations. The policy network is implemented as a Deep Sets architecture (Zaheer et al., 2017) that maps node features to actions with an MLP. Learnable weights are shared among nodes.

Additional results including comparisons with a non-modular baseline are reported in Appendix C.2. Tab. 1 provides a summary of the salient properties of each architecture. Additional details are reported in Appendix D. For FGNN, message-passing layers simply propagate representations among neighbors and not across different levels; messages flow bottom-up at the encoding step only. In models exploiting the hierarchical structure, the intrinsic re-

Table 1: Summary of baselines and variants used.

|  | FGNN | FDS | GNN | DS |
|---|---|---|---|---|
| Hierarchy | ✓ | ✓ | ✗ | ✗ |
| Message passing | ✓ | ✗ | ✓ | ✗ |
| Modularity | ✓ | ✓ | ✓ | ✓ |

wards of lower levels incentivize behaviors that align nodes with the assigned goals. More precisely, at each time step, the intrinsic signal of a node is defined as a function of the cosine similarity between its states and the received commands (refer to Appendix D.2 for a detailed explanation).

**Optimization algorithm** In FGRL, each graph $\mathcal{G}_{l_h}$ represents a structured agent in which each node acts according to a shared policy that is specific for that level. Each level is trained independently from the others as it maximizes its own reward signal. In principle, the model could also be trained end-to-end, i.e., as a single network, but this would most likely result in the collapse of goals' representations, and would not allow for having different reward signals for each level. We provide a sensitivity analysis of this multi-level optimization approach in Appendix C.3.

In principle, each level-specific policy can be paired with a standard RL optimization algorithm, but the hierarchical paradigm introduces additional challenges in the training procedure. As an example, instabilities at a single level can hinder global performance and make credit assignment more difficult: good commands propagated by the upper layers are not rewarded properly if workers fail to select the correct actions. We use *Covariance Matrix Adaptation Evolution Strategy* (CMA-ES) (Hansen & Ostermeier, 2001) as the optimization method since evolutionary algorithms have proven to be competitive with policy gradient methods and less likely to get stuck in such sub-optimal solutions (Salimans et al., 2017); we provide a comparison with a standard gradient-based algorithm in Appendix C.2.

Finally, policies of intermediate levels can commit to sub-optimal behaviors if they receive the intrinsic reward as the only learning signal. Therefore, similarly to Vezhnevets et al. (2017), we add the external reward to the intrinsic one at each level. Further details regarding both the policy optimization and reward scheme can be found in Appendix D.1 and D.2, respectively.

## 5.2 Graph Clustering Problem

Given a graph with $\beta$ communities and $N_\beta$ nodes per community with random labels, the objective is to cluster nodes so that only nodes belonging to the same community have the same label, i.e., are assigned to the same cluster. The agents receive a sparse reward based on the *MinCut loss* (Bianchi et al., 2020) at the end of each episode, with an additional reward bonus for eventually solving the problem. Further details regarding observation space, action space, and reward function can be found in Appendix B. The hierarchical graph $\mathcal{G}^*$ of both FGNN and FDS is a 3-level graph that is built by assigning a sub-manager to each community; all the $\beta$ sub-managers are then connected to a single top-level manager. As a result, goals sent to workers and sub-managers are conditioned to be a $\beta$-dimensional vector representing a target assignment at the node level and community level, respectively. Furthermore, we let the top-level manager and sub-managers act on different time scales by keeping the selected goals fixed for 10 and 5 time steps, respectively. All baselines have access to static coordinates $\boldsymbol{f} \in \mathbb{R}^2$ of each node.

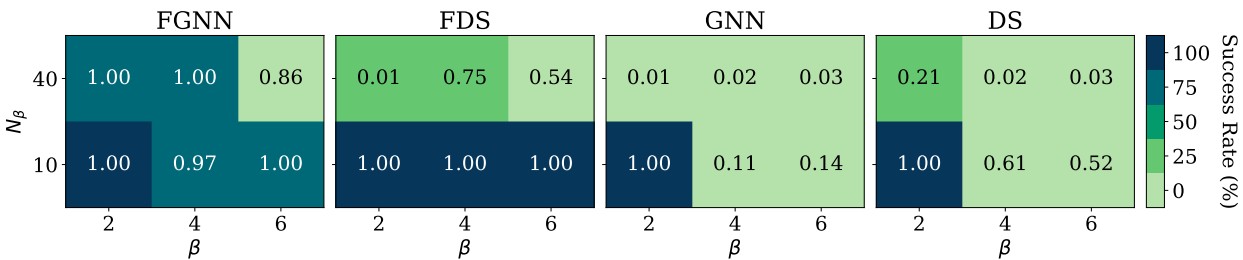

Figure 3: Percentage of correct clustering (color) and median of NMI (value) over 4 independent runs. We remark that given a configuration $(\beta, N_\beta)$, all the models are trained on the same topology to ensure fairness.

In Fig. 3 we show the success rate of each agent in clustering the graph and the median of the Normalized Mutual Information (NMI) score computed across different runs. Further details regarding the computation of the NMI are reported in Appendix C.1. FGNN outperforms all the baselines in settings with large communities, i.e., with $N_\beta = 40$: it fails to perfectly solve the task only in the environment with the highest complexity, i.e., with $\beta = 6$ and $N_\beta = 40$, where the corresponding NMI score indicates a good clustering nonetheless. The performance of the FGRL model without message passing (FDS) deteriorates when the number of nodes per community increases, while it is able to perfectly solve the task for $N_\beta = 10$. Indeed, for such small communities, static coordinates alone appear sufficient for the feudal paradigm to reach one of the possible target configurations, while message-passing features result in being redundant. The comparison of the clustering scores achieved by the modular baselines without hierarchy for $N_\beta = 10$ further supports this observation: the DS model, which is not graph-based, achieves better NMI scores than GNN. In general, modular baselines can solve simple scenarios, but performance degrades rapidly as the complexity increases. When compared with GNN, results obtained by FGNN support our claims: the hierarchical approach effectively enables long-range coordination. However, a hierarchical architecture alone (without any local message passing) fails in solving the task for large communities where message-passing helps in coordinating the decision-making process locally.

**Effect of temporally extended goals** As already mentioned, the top-level manager and sub-managers send a new command every 10 and 5 steps, respectively. By doing so, in the 10-step interval sub-managers set 2 commands, i.e., one every 5 steps, to steer subordinate workers towards a specific configuration; in turn, each worker has 5 time step to reach its assigned label configuration and obtain a positive intrinsic reward. We investigate the effectiveness of temporally extended goals by comparing the results of FGNN and FDS against versions of these models where (sub-)managers send a different goal at each time step (one-step goals). Results in Fig. 4 show that this mechanism is pivotal for solving the task. In particular, while the performance of FGNN with $N_\beta = 10$ nodes is only slightly worse with one-step goals, the upper levels in the hierarchy fail to properly coordinate levels below for $N_\beta = 40$, resulting in a drastic performance decrease. FDS with long-term goals outperforms its one-step counterpart when $N_\beta = 10$; however, none of the FDS variants is able to achieve good results for $N_\beta = 40$.

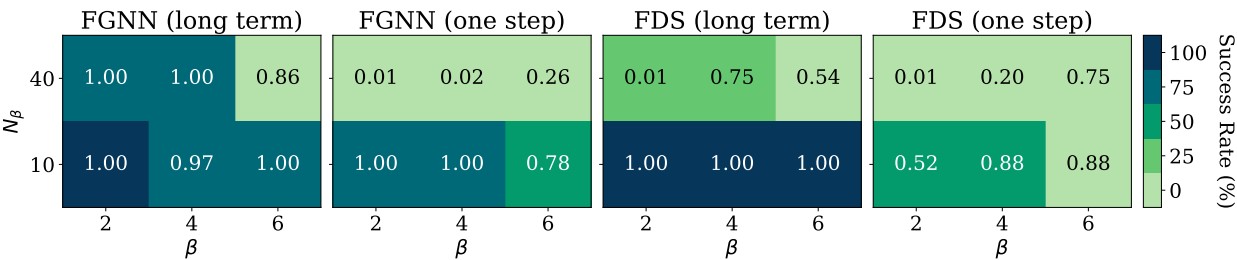

Figure 4: Percentage of correct clustering (color) and median of NMI score (value) over 4 independent runs with long-term and one-step goals. We remark that given a configuration $(\beta, N_\beta)$, all the models are trained on the same topology to ensure fairness.

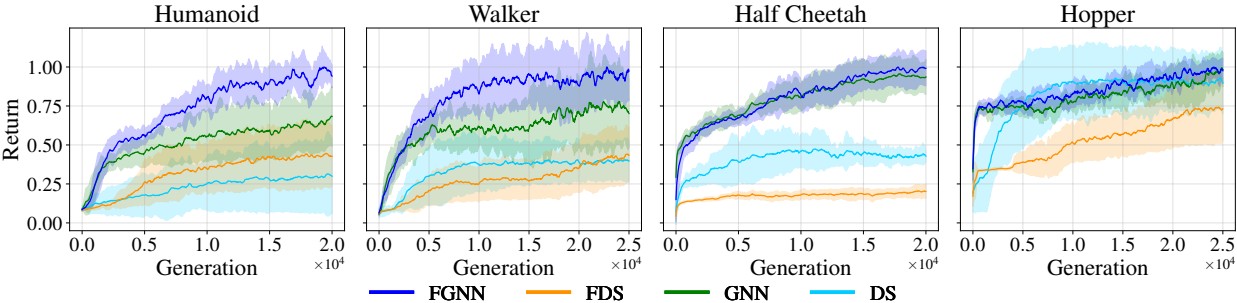

Figure 5: Average return and standard deviation of the considered agents on the MuJoCo benchmarks (averaged over 4 runs). Each generation refers to a population of 64 episodes. To ease the visualization, the plots show a running average with returns normalized w.r.t. the maximum obtained values, that are: 4025 (Humanoid), 2817 (Walker), 1918 (Half Cheetah), and 3175 (Hopper).

### 5.3 MuJoCo Benchmarks

We consider the modular variant of 4 environments adapted from the standard MuJoCo locomotion tasks – namely 'Humanoid', 'Walker2D', 'Hopper', and 'Half Cheetah'. For these variants, we use the same setup introduced by Huang et al. (2020): differently from the original environments where states (actions) are global, here the state (action) space is a collection of local limb states (actions), one for each actuator of the agent. The composable nature of such variants makes GNNs suitable candidates for tackling these problems (Wang et al., 2018; Huang et al., 2020). In particular, the rationale for selecting these environments as benchmarks is that empirical evidence (Kurin et al., 2020) shows the limitations of flat GNNs in achieving good coordination and motivates the adoption of our hierarchical approach in these settings. If agents do not crash, episodes last for 1000 time steps; environment rewards are defined according to the distance covered: the faster the agent, the higher the reward. Nodes in the base graph are the actuators of the physical agent: hence, in hierarchical models (FGNN and FDS), workers are clustered using a simple heuristic, i.e., grouping together nodes belonging to the same body part (see Appendix D.3 for details).

We report the results for the 4 agents in Fig. 5. Returns obtained by the FGNN architecture in 'Humanoid' and 'Walker' support the adoption of message passing within the feudal paradigm in structured environments, where the agent requires a higher degree of coordination. Indeed, hierarchical graph-based policies outperform the baselines by a larger margin in the more complex tasks. In 'Half Cheetah', FGNN and GNN achieve similar results. A possible explanation for this behavior can be found by considering the morphology of the agent itself: this agent cannot fall, i.e., there is no need for high-level planning accounting to maintain balance. Similarly, in 'Hopper' (i.e., the agent with the simplest morphology), good locomotion policies can be learned without a high degree of coordination, making the architecture of the more sophisticated models redundant. As one would expect, then, the performance of FGNN and GNN for this agent is comparable with that of DS, i.e., the simpler modular variant. In general, FDS obtains subpar performance across environments, suggesting that, in modular architecture, the feudal paradigm might not be as effective without any other coordination mechanism.

**Analysis of generated goals** The feudal paradigm relies on the goal-generation mechanism, that establishes communication among different hierarchical levels by propagating goals. This key feature enables policy improvement on both a global and local scale: the top-level manager has to generate goals aligned with the external return, while sub-managers must learn to break down the received goals into subtasks.

To investigate whether commands are generated coherently, we run t-SNE (Van der Maaten & Hinton, 2008) on goal vectors received by pairs of nodes with symmetric roles in the morphology of a trained 'Walker' agent and analyze their time evolution during an episode. More precisely, for each pair of nodes (e.g., left and right knee) we collect the goals signals received in an episode, compute the one-dimensional t-SNE embeddings from such sample, and then plot the embeddings corresponding to the goals sent to the left and right nodes.

Empirical observations emerging from the results in Fig. 6 provide experimental evidence to support the effectiveness of the proposed goal-propagation mechanism. First, we highlight that commands sent by the top-level manager to the intermediate sub-managers responsible for legs (purple and blue nodes in Fig. 10) show a temporal structure that is coherent with the oscillating patterns of worker nodes. Indeed, goals are generated using a top-down approach (refer to Eq. 4) where commands assigned to lower levels are a function of those received by upper levels; as such, the temporal structure is preserved. Second, while curves associated with workers intersect at time steps corresponding to the actual steps taken by the physical agent, goal representations received by left and right sub-managers appear less predictable, and the corresponding t-SNE embeddings are more diverse. Indeed, the abstraction of commands increases with the depth of the hierarchy: goals at the upper levels are not directly mapped into physical actions and should provide supervision to the levels below. Conversely, goals assigned by sub-managers to workers become more specific and, e.g., depend on the specific joints being controlled. This analysis shows that propagated goals are meaningful and capture salient aspects of the learned gait. Results show coordination emerging from the interaction of nodes at different levels and support the adoption of such an architecture to implement hierarchical control policies.

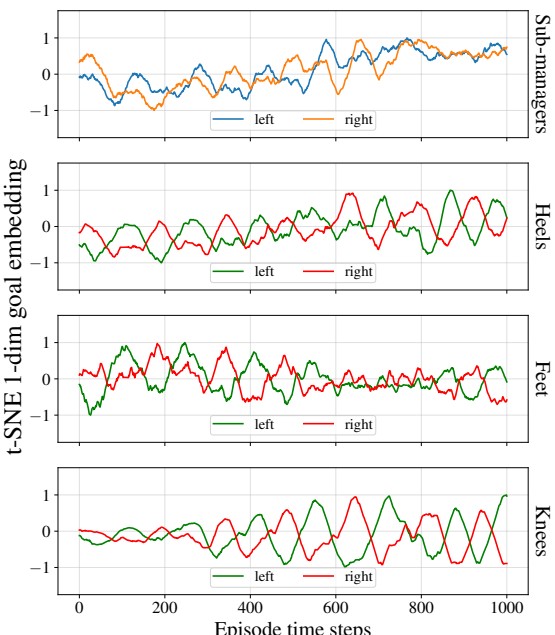

Figure 6: Analysis of goals received by sub-managers (orange and blue lines) and workers (green and red lines) in an episode of a 'Walker' environment (refer to Fig. 10 for the hierarchical graph). Plots show a 50-step running average.

## 6 Conclusions and Future Works

We proposed a novel hierarchical graph-based reinforcement learning framework, named *Feudal Graph Reinforcement Learning*. Our approach exploits the feudal RL paradigm to learn modular hierarchical policies within a message-passing architecture. We argue that hierarchical graph neural networks provide the proper computational and learning framework to achieve spatiotemporal abstraction. In FGRL, nodes are organized in a multilayered graph structure and act according to a committee of composable policies, each with a specific role within the hierarchy. Nodes at the lowest level (workers) take actions in the environment, while (sub-)managers implement higher-level functionalities and provide commands at levels below, given a coarser state representation. Experiments on graph clustering and MuJoCo locomotion benchmarks – together with the in-depth analysis of the learned behaviors – highlight the effectiveness of the approach.

There are several possible directions for future research. In the first place, the main current limitation of FGRL, shared among several HRL algorithms (Vezhnevets et al., 2017), is in the inherent issues in jointly and efficiently learning the different components of the hierarchy in an end-to-end fashion. Solving this limitation would facilitate learning hierarchies of policies less reliant on the external reward, allowing state-of-the-art optimization methods to be integrated into the framework while ensuring stability in the training process. In this regard, different implementations of the intrinsic reward mechanism could be explored. Lastly, while in this paper we addressed limitations of flat graph-based approaches, future directions might investigate the effectiveness of the proposed framework in scenarios where credit assignment is more challenging. Preliminary experiments conducted in Sec. 5.2 show promising results in this sense.

## Acknowledgements

This research was funded by the Swiss National Science Foundation under grant 204061: High-Order Relations and Dynamics in Graph Neural Networks.

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

## Appendix

## A    Extraction of the Hierarchical Graph

In Fig. 7 we report an example of extraction of the hierarchical graph $\mathcal{G}^*$ for the 'Humanoid' environment; the base graph $\mathcal{G}_1$ is obtained directly from the agent's morphology (see Fig. 1 for reference) and the number of layers of $\mathcal{G}^*$ is a hyperparameter.

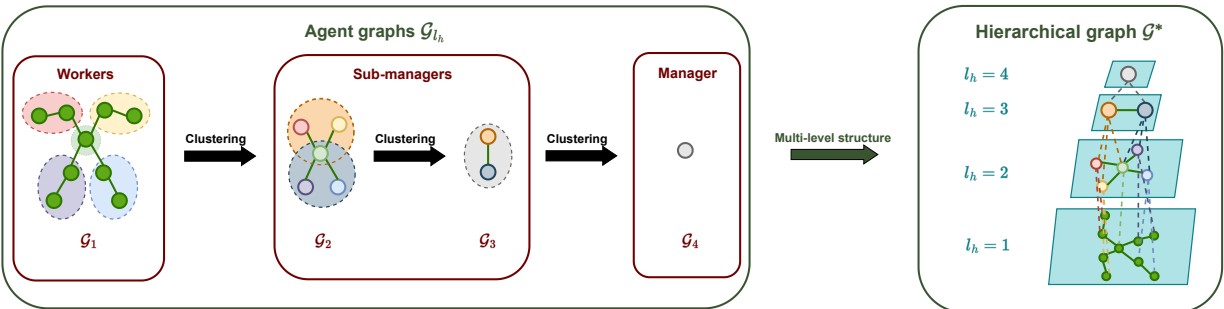

Figure 7: Extraction of the hierarchical graph $\mathcal{G}^*$ starting from the workers' graph $\mathcal{G}_1$ of the 'Humanoid' environment. Hierarchical edges of $\mathcal{G}^*$ are represented as dashed lines and denote parent-child relationships. As shown in the second level, each node can have more supervisors.

## B    Graph Clustering Environment

The environment of the synthetic graph clustering problem is defined by a $\beta$-community graph with $N_\beta$ nodes for each community, that is, a community graph with $\beta N_\beta$ nodes. The objective is to cluster the graph so that each community has a unique label for all the nodes, different from other communities. For each simulation, we generate a $(\beta, N_\beta)$ graph with binary adjacency matrix $\mathbf{A} \in \{0,1\}^{\beta N_\beta \times \beta N_\beta}$ and degree matrix $\mathbf{D} = \mathrm{diag}(\mathbf{A1}_N)$, under the assumption that there are no isolated nodes. Each node $i$ has a static feature vector given by the $\boldsymbol{f}_i \in \mathbb{R}^2$ coordinates: nodes belonging to the same community are characterized by similar features. At each time step $t$, the observation vector $\boldsymbol{x}_i^t$ of each $i$-th node is the concatenation of the one-hot encoding of the current cluster assignment $\boldsymbol{s}_i^t$, its coordinates, and the normalized remaining time steps before the end of the episode:

$$\boldsymbol{x}_i^t = \boldsymbol{s}_i^t \,||\, \boldsymbol{f}_i \,||\, 1 - t/T \tag{7}$$

We remark that assignments are initialized at random at the beginning of each episode. The action is a 3-dimensional vector *(left/no-op/right)* that allows nodes to change cluster by possibly changing its label of at most one index; periodic boundary conditions are applied on the label vector. We consider an episodic setting where episodes last for $T = 50$ time steps, unless the graph reaches one of the possible target configurations where nodes belonging to the same community have the same label. The sparse reward at the termination time step $t \leq T$ is given by:

$$R_t = \frac{Tr(\mathbf{C}^T \tilde{\mathbf{A}} \mathbf{C})}{Tr(\mathbf{C}^T \tilde{\mathbf{D}} \mathbf{C})} - \left\| \frac{\mathbf{C}^T \mathbf{C}}{\|\mathbf{C}^T \mathbf{C}\|_F} - \frac{\mathbf{I}_K}{\sqrt{K}} \right\|_F + (T - t) \tag{8}$$

where $\mathbf{C}$ is the cluster matrix, $\tilde{\mathbf{A}} = \mathbf{D}^{-\frac{1}{2}} \mathbf{A} \mathbf{D}^{-\frac{1}{2}} \in \mathbb{R}^{\beta N_\beta \times \beta N_\beta}$ is the normalized adjacency matrix, and $\tilde{\mathbf{D}}$ is the degree matrix of $\tilde{\mathbf{A}}$. The first two terms represent the negative of the *MinCut loss* (Bianchi et al., 2020): the first one promotes clustering among strongly connected nodes, while the second one prevents degenerate solutions by ensuring similarity in size and orthogonality between different clusters. We remark that, for this maximization problem, those terms are bounded in $[0, 1]$ and $[-2, 0]$, respectively: therefore, the bonus factor $(T - t)$ strongly encourages solving the task with as few iterations as possible. We consider the task as solved if the running average of the success rate percentages of the last 20 evaluation samples is greater than 95%.

# C   Additional Results

## C.1   NMI Score in the Graph Clustering Problem

The NMI score measures the similarity between two independent label assignments $X, Y$. It is computed as:

$$NMI(X,Y) = \frac{2I(X;Y)}{H(X) + H(Y)}, \tag{9}$$

where $H(\cdot)$ and $I(X;Y)$ denote the entropy of the labels and mutual information, respectively. This metric is bounded in $[0, 1]$, where the boundaries represent perfect dissimilarity ($NMI = 0$) and similarity ($NMI = 1$), and is invariant under labels permutation. Hence, in our setting, we measure the NMI score of the predicted clustering w.r.t. a generic target configuration where nodes belonging to the same community have the same cluster-specific label. In Tab. 2 we report the maximum (blue) and minimum (red) values achieved by the models in each configuration (Fig. 3 and 4 for reference).

Table 2: Minimum and maximum values of the NMI score achieved by the models in the graph clustering problem.

| | $N_\beta = 10$ | | | $N_\beta = 40$ | | |
|---|---|---|---|---|---|---|
| | $\beta = 2$ | $\beta = 4$ | $\beta = 6$ | $\beta = 2$ | $\beta = 4$ | $\beta = 6$ |
| FGNN | (1.00, 1.00) | (0.10, 1.00) | (0.15, 1.00) | (0.01, 1.00) | (0.02, 1.00) | (0.84, 0.94) |
| FDS | (1.00, 1.00) | (1.00, 1.00) | (1.00, 1.00) | (0.01, 1.00) | (0.02, 1.00) | (0.39, 1.00) |
| GNN | (1.00, 1.00) | (0.11, 0.24) | (0.14, 0.15) | (0.01, 0.03) | (0.02, 0.02) | (0.03, 0.03) |
| DS | (1.00, 1.00) | (0.21, 0.76) | (0.42, 0.88) | (0.01, 1.00) | (0.02, 0.03) | (0.03, 0.03) |
| FGNN (one-step goals) | (0.03, 1.00) | (0.09, 1.00) | (0.16, 1.00) | (0.01, 0.02) | (0.02, 0.88) | (0.03, 0.79) |
| FDS (one-step goals) | (0.04, 1.00) | (0.76, 1.00) | (0.76, 0.88) | (0.004, 0.013) | (0.02, 0.44) | (0.60, 0.87) |

## C.2   Comparison with Non-modular Baselines

The FGRL framework is specifically designed for learning modular policies. Therefore, the experiments conducted in Sec. 5.2 and 5.3 validate the effectiveness of the full proposed methodology (FGNN) in comparison to other modular approaches, such as *flat* GNNs. Nevertheless, to better contextualize the results, here we provide a comparison with an MLP (a non-modular policy) where each state and action are the concatenation of node-level observations and actions, respectively. Being non-modular, it cannot be directly applied to different morphologies and the model size increases with the number of nodes. We remark that, in Sec. 5.2 and 5.3, all the models were trained using the same learning algorithm, i.e., CMA-ES (Hansen & Ostermeier, 2001), to highlight the performance differences brought by each architectural variant rather than those coming from the learning algorithm. Specifically, we chose evolution strategies because, as mentioned in Sec. 5.1, learning a hierarchy of level-specific policies using independent reward signals is challenging. In this regard, this section provides a comparison against two different MLP baselines that use different learning algorithms: one is trained with CMA-ES to assess the difference in performance w.r.t. the composable FGNN, while the other uses *Proximal Policy Optimization* (PPO) (Schulman et al., 2017) and allows us to analyze the results in the broader context of gradient-based methods. For simplicity, we denote the two variants as MLP and PPO, respectively. Hyperparameters are reported in Appendix D.4 and the code for PPO was adapted from a public repository (Barhate, 2021). We chose PPO as a gradient-based algorithm because of its popularity and wide applicability in both single and multi-agent reinforcement learning problems (Yu et al., 2022).

In general, evolution strategies rely on a selection process that evaluates a sample (population) of parameters for each update step, resulting in worse sample efficiency compared to gradient-based methods that can

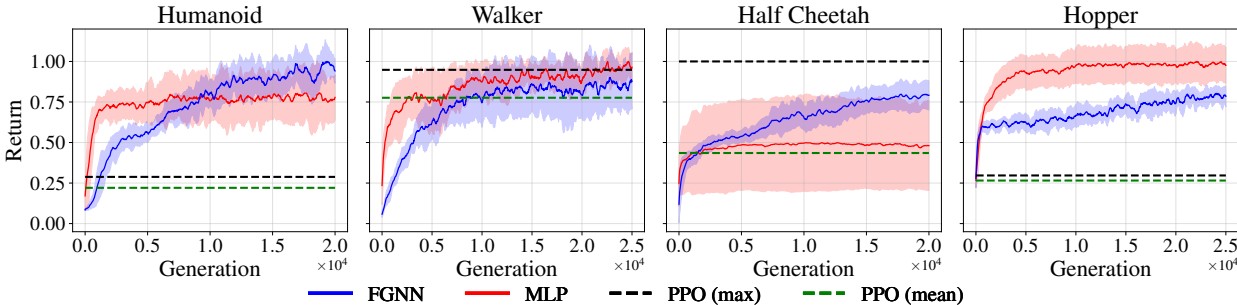

Figure 8: Comparison of the average return achieved with FGNN (4 runs) and MLP (4 runs) with average and maximum value of PPO (5 runs) at the last training step in the 4 MuJoCo benchmarks. To ease the visualization, the plots show a running average with returns normalized w.r.t. the maximum obtained values, that are: 4025 (Humanoid), 3125 (Walker), 2402 (Half Cheetah), and 3950 (Hopper).

instead perform multiple updates with a smaller sample. Therefore, in Fig. 8 we report the average returns obtained with FGNN and MLP, together with the mean and maximum values among 5 independent seeds of PPO at the last training step, i.e., $2 \cdot 10^7$; we applied a running average to account for fluctuations. PPO hyperparameters were tuned on the 'Walker' agent (as we similarly did for CMA-ES). In this scenario, the performance of FGNN and MLP is comparable with PPO, highlighting that evolution strategies allow to achieve absolute returns that are competitive w.r.t. a widely used gradient-based method. Conversely, PPO appears to be less robust when the same hyperparameters are used to learn locomotion policies for other morphologies. FGNN performs similarly or better than the MLP baseline in all the environments, except for 'Hopper': as already mentioned, this agent has the simplest morphology and, as expected, a simple MLP is enough to learn a good policy. Furthermore, we qualitatively observed that FGNN learns more natural gaits in 'Humanoid' and 'Walker'.

### C.3 Analysis of the Multi-Level Optimization

The proposed feudal framework relies on a multi-level hierarchy where nodes at each level $\mathcal{G}_{l_h}$ act according to a level-specific policy trained to maximize its own reward. Notably, the policy trained at level $l_h$ completely ignores rewards received at different levels: coordination completely relies on the message-passing and goal-generation mechanisms. This peculiar aspect implies that each policy can have its own policy optimization routine.

To analyze the impact of such a multi-level structure and optimization routine, we compare results obtained by a 3-level FGNN model in the 'Walker' environment against 1) a 3-level FGNN model without intrinsic reward, 2) a 2-level FGNN model, and 3) a 3-level FGNN model where all the policies are jointly trained to maximize the external reward only. We remark that the number of learnable parameters is the same for all the 3-level variants. As shown in Fig. 9, our method achieves the highest average return, while baselines are more likely to get stuck in sub-optimal policies (e.g., by learning to jump rather than to walk). Comparing the full model (blue curve) with the variant without intrinsic reward (left, red curve) highlights the advantage of the hierarchical intrinsic reward mechanism. In the full 3-level FGRL model, the reward at each level incentivizes workers and sub-managers to act accordingly to the received commands. This auxiliary task turns out to be instrumental in avoiding sub-optimal gaits. The 2-level model (middle, green curve) performs akin to the variant without intrinsic reward, hence hinting at the benefits of using a deeper hierarchy for agents with complex morphologies. Lastly, for the variant where all levels are jointly optimized (right, black curve), empirical results show that the resulting agent does not only achieve a lower return but, surprisingly, it is also less sample efficient.

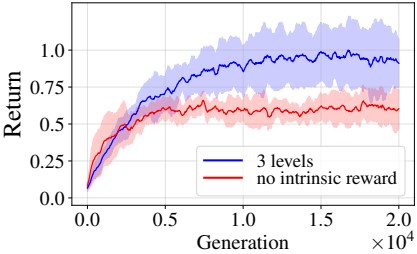 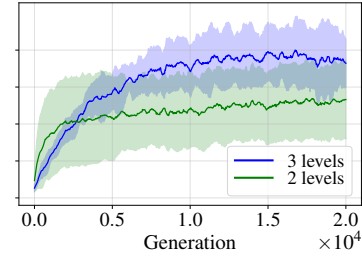 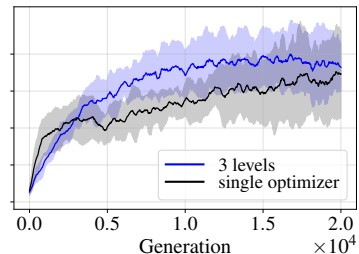

Figure 9: Analysis of the 3-level FGNN model with variants that use a different reward choice (left), depth of the hierarchy (middle), and optimization scheme (right) in a 'Walker' environment (4 seeds for each variant). Returns are reported with standard deviation and each generation refers to a population of 64 episodes. To ease the visualization, the plot shows a running average with returns normalized w.r.t. the maximum obtained value, i.e., 2725.

# D  Implementation Details

## D.1  Optimization Algorithm

As already mentioned, to learn the parameters for our setup we use evolution strategies, namely CMA-ES (Hansen & Ostermeier, 2001). The number of instances of CMA-ES we initialize corresponds to the depth of the hierarchy, as each level has its independent policy. Indeed, each CMA-ES optimizes only modules corresponding to its level:

- Workers CMA-ES (level 1): weight matrix $\boldsymbol{W}_1$, functions $\mu$ and $\phi_1^1$.

- Sub-managers CMA-ES (levels $l_h \in \{2, \ldots, L_h - 1\}$): functions $\rho^{l_h}$, $\psi^{l_h}$, and $\phi_1^{l_h}$.

- Manager CMA-ES (level $L_h$): functions $\rho^{L_h}$ and $\psi^{L_h}$.

Experiments were run on a workstation equipped with AMD EPYC 7513 CPUs. Depending on the model and environment, each seed can take from 30 minutes to 1 day for the graph clustering experiment and from 12 hours to 3 days for the MuJoCo benchmarks. Code to reproduce experiments is available online at https://github.com/tommasomarzi/fgrl.

## D.2  Intrinsic Reward

In variants that take advantage of the hierarchical structure (FGNN and FDS), we define the intrinsic reward of sub-managers and workers as a signal that measures their alignment with the assigned goals. At each time step $t$, for each hierarchical level we add the environment reward to the average intrinsic signal. For both environments, the reward scheme was inspired by Vezhnevets et al. (2017).

**Graph clustering problem**  In this environment, each $i$-th node has a single (sub-)manager $\mathcal{P}(i)$ and propagated goals represent a target label. At each time step, the reward of each $i$-th worker is defined as:

$$r_i = \frac{1}{T}\Big[d_c\left(\boldsymbol{g}_{\mathcal{P}(i)\to i}, \boldsymbol{s}_i'\right) - 0.5\Big], \tag{10}$$

where $d_c$ is the cosine similarity function, while $\boldsymbol{g}_{\mathcal{P}(i)\to i}$ and $\boldsymbol{s}_i'$ denote the one-hot encoding of the received command and subsequent cluster, respectively. Similarly, intermediate nodes are responsible of the aggregated cluster of subordinated nodes, and the intrinsic reward of each $i$-th sub-manager can be defined as:

$$r_i = \frac{1}{T}\left[d_c\left(\boldsymbol{g}_{\mathcal{P}(i)\to i}, \sum_{k\in\mathcal{C}(i)}\frac{\boldsymbol{s}_k'}{|\mathcal{C}(i)|}\right) - 0.5\right], \tag{11}$$

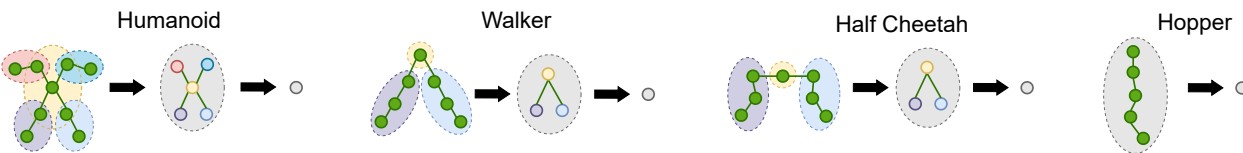

Figure 10: Extraction of the hierarchical graphs $\mathcal{G}^*$ used in the FGNN model. The filled colored ellipses with dashed lines highlight the hierarchical connections among different levels, and overlapping circles imply that the node is subordinate of both the sub-managers. Graphs with green nodes represent the original agent graphs $\mathcal{G}_1$.

Since $\boldsymbol{s}'_k$ is a one-hot encoding vector and goals represent a target label, we set the one-step intrinsic rewards in the range $r_i \in [-0.01, 0.01]$ to penalize subordinate nodes for not following assigned commands. In order to be comparable with the values of the *MinCut loss*, intrinsic signals are normalized w.r.t. the length of the episode.

**MuJoCo benchmarks** In this control problem, goals are instead unconstrained latent vectors. Nodes at the lowest level have the most fine-grained representations and learn to follow goals at the simulation scale. Thus, each $i$-th worker receives an intrinsic signal:

$$r_i = d_c \left( \operatorname*{AGGR}_{j \in \mathcal{P}(i)} \{\boldsymbol{g}_{j \to i}\}, \boldsymbol{s}'_i - \boldsymbol{s}_i \right) + 1, \tag{12}$$

where $\boldsymbol{s}'_i$ denotes the subsequent state. Conversely, intermediate nodes operate on a coarser scale, and the intrinsic reward of each $i$-th sub-manager can be similarly defined as:

$$r_i = d_c \left( \operatorname*{AGGR}_{j \in \mathcal{P}(i)} \{\boldsymbol{g}_{j \to i}\}, \boldsymbol{h}_i^{0\prime} \right) + 1, \tag{13}$$

where $\boldsymbol{h}_i^{0\prime}$ denotes the subsequent initial state representation, computed using Eq. 2. Since goals are latent, we prevent lower levels from getting a negative reward for not following the assigned commands in the first stages of learning by providing positive intrinsic rewards in the range $r_i \in [0, 2]$.

### D.3 Hierarchical Graphs

The graphs under analysis for the MuJoCo benchmarks have a low number of nodes, and each actuator has a proper physical meaning in the morphology of the agent. Thus, we decide to create the hierarchical graphs using heuristic. As an example, in the 'Walker' agent we expect nodes of the same leg to be clustered together, and the associated sub-manager to be at the same hierarchical level as that of the other leg: in this way, the topology of the intermediate graph reflects the symmetry of the agent graph $\mathcal{G}_1$.

The hierarchical graphs for the FGNN model are reported in Fig. 10. Notice that the 'Hopper' agent has a simple morphology with no immediate hierarchical abstraction and where each actuator has a different role: as a consequence, a meaningful hierarchy cannot be trivially extracted, and results revealed no benefit in implementing a 3-level hierarchy for this agent. We remark that hierarchical graphs used in the FDS variant are not reported because in all the environments empirical evidence did not show improvements as the depth of the hierarchy increased, leading to 2-level hierarchical graphs where all the workers are connected to a single top-level manager (see 'Hopper' in Fig. 10 for an example).

### D.4 Reproducibility

The code for the experiments was developed by relying on open-source libraries[1] and on the publicly available code of previous works [2,3]. All the learnable components in message-passing blocks are implemented as

---

[1]ESTool: `https://github.com/hardmaru/estool`
[2]SMP Huang et al. (2020): `https://github.com/huangwl18/modular-rl`
[3]NerveNet Wang et al. (2018): `https://github.com/WilsonWangTHU/NerveNet`

MLPs. In Tab. 3 are reported the values of the hyperparameters used in our experiments. For each baseline, we tuned the number of hidden units and CMA-ES step size by performing a grid search on the average return. Note that for the MuJoCo benchmarks the best configuration of the MLP baseline and that of FGNN results in a similar number of learnable parameters. For the graph clustering experiment we used the same set of hyperparameters, except for the aggregation function of subordinate nodes in the feudal models where we used the average instead of the sum.

The hyperparameters used for the comparison with PPO (Appendix C.2) are the following: we used Adam optimizer (Kingma & Ba, 2014) with a learning rate of $3 \cdot 10^{-6}$ and hidden layers $[64, 64]$ with *tanh* non-linearities for both actor and critic; discount factor $\gamma$ and clipping value $\epsilon$ are fixed to 0.99 and 0.2, respectively; policy is updated for 10 epochs with batch size of 64 and the updating horizon is 2048 time steps; the initial action standard deviation is 0.6 and it decays every $10^4$ episodes of 0.05, up to a minimum of 0.2. We performed a grid search on such hyperparameters, focusing mainly on learning rate, hidden layers, updating epochs, and updating horizon.

Table 3: Hyperparameters used in each model, where the ✗ marker indicates those that are not part of the architecture. For the total number of parameters, in the non-modular baseline (MLP) we reported the maximum among the four environments of the MuJoCo benchmark. We remark that in the modular models, the number of parameters does not depend on the environment, but since in the feudal architectures it depends on the hierarchical height, in FGNN we reported the number corresponding to the maximum one, i.e., 3 levels.

| Context | Hyperparameter | MLP | DS | GNN | FDS | FGNN |
|---------|----------------|-----|-----|------|------|------|
| CMA-ES | Population size | 64 | 64 | 64 | 64 | 64 |
|  | Initial step size | 0.25 | 0.25 | 0.25 | 0.5 | 0.25 |
| Policy | Dimension of state representation | ✗ | ✗ | 32 | 32 | 20 |
|  | Dimension of hidden layer | 64 | 32 | 32 | 32 | 30 |
|  | Activation function | tanh | tanh | tanh | tanh | tanh |
|  | AGGR (subordinate nodes) | ✗ | ✗ | sum | ✗ | sum |
|  | AGGR (message passing) | ✗ | ✗ | sum | ✗ | sum |
|  | AGGR (goals aggregation) | ✗ | ✗ | ✗ | mean | mean |
|  | Maximal hierarchy height | ✗ | ✗ | ✗ | 2 | 3 |
|  | Message-passing rounds | ✗ | ✗ | 2 | ✗ | 2 |
|  | Shared weights (message passing) | ✗ | ✗ | ✓ | ✗ | ✓ |
|  | # of parameters (MuJoCo benchmark) | 11593 | 673 | 4865 | 7124 | 12700 |

