# OpenReview forum: "Feudal Graph Reinforcement Learning"
_TMLR — Accepted by TMLR_

### Review · Reviewer_14WH · 2024-10-20

**Summary Of Contributions:**

- Proposes a graph neural network -based hierarchical RL method.
- Evaluates on MuJoCo and graph clustering tasks.

**Audience:**

Yes

**Claims And Evidence:**

Yes

**Requested Changes:**

- Since Kurin et al. [2020], a number of other works targeting similar graph-based morphology control setting have been published. These seem like missing related work from the current paper. Discussion of similarities should be included.
	- Trabucco et al. 2022., "AnyMorph: Learning Transferable Polices By Inferring Agent Morphology".
	- Gupta et al., 2022., "Metamorph: Learning universal controllers with transformers".
	- Xiong et al., 2023., "Universal Morphology Control via Contextual Modulation".
	- Chen et al., 2023., "Subequivariant Graph Reinforcement Learning in 3D Environments".
	- Xiong et al., 2024., "Distilling Morphology-Conditioned Hypernetworks for Efficient Universal Morphology Control".
- Since the whole architecture depends on the submanagers and workers optimizing their rewards in a useful way, relegating the discussion of the reward design fully to the appendix seems odd. Please include a high-level description of the reward you ended up using in the main paper.
- Apparently the MuJoCo setup introduced by Huang et al. [2020] is different from the vanilla MuJoCo setting considered elsewhere in RL. Please include a brief description on how the environments differ and why standard "flat" (as opposed to hierarchical) RL is not appropriate.
- The discussion of the temporal structure in Figure 6 is not clear. I don't think you can say that the top panel (sub-managers) has less structure in time than the third panel (feet). Furthermore, whatever temporal structure there is in the top panel seems highly correlated with the structure in the other panels. That said, I like this analysis, please make it clearer.
- While this is a well presented and interesting approach to HRL, the paper does not really explore the limitations of HRL in general and this method specifically. There is a sentence referring to the limitations listed in Vezhnevets et al. [2017] but one would imagine that research in HRL would have a different set of concerns more than seven years later. It does not become clear from the manuscript how applicable the proposed method is, e.g., in standard RL environments with challenging temporal credit assignment problems, for which HRL has been proposed as a solution. Discussion of the limitations would help place this work in broader context.

**Strengths And Weaknesses:**

- Graph neural networks and feudal RL seem like a reasonably good match. It is a valuable contribution to the community to explore these together.
- The paper is well written and has clear figures illustrating the concepts.
- Some details are missing or are not clearly presented in the current manuscript. Please see requested changes for more.

---

> ### Author Response · Authors · 2024-10-31
>
> Thank you for the detailed review, please find our point-by-point answers below.
>
> > Since Kurin et al. [2020], a number of other works targeting similar graph-based morphology control setting have been published. These seem like missing related work from the current paper. Discussion of similarities should be included.
>
> Thank you for the references, we will add them to the main text.
>
> > Since the whole architecture depends on the submanagers and workers optimizing their rewards in a useful way, relegating the discussion of the reward design fully to the appendix seems odd. Please include a high-level description of the reward you ended up using in the main paper.
>
> Thank you for the comment, we agree with the reviewer that some details should be discussed in the main paper. We will add the discussion on the intrinsic reward scheme in Sec. 5.1.
>
> > Apparently the MuJoCo setup introduced by Huang et al. [2020] is different from the vanilla MuJoCo setting considered elsewhere in RL. Please include a brief description on how the environments differ and why standard "flat" (as opposed to hierarchical) RL is not appropriate.
>
> The main difference is that state (action) space is factored in a set of observation vectors (actions) associated with each limb (actuator). According to the MuJoCo version, there could also be differences in the physical simulator and available observations. The main reason why we chose these environments is that they are popular benchmarks for composable RL methods [1,2]. Furthermore, while flat GNNs can obtain reasonably good results in these benchmarks, Kurin et al. [3] showed that bottlenecks in the flat message-passing architecture could hinder performance, thus motivating the adoption of our hierarchical approach.
>
> > The discussion of the temporal structure in Figure 6 is not clear. [...]
>
> Admittedly, the discussion can be improved and might have resulted in misunderstanding. We agree with the reviewer that the temporal structure of the top panel is correlated with the other panels, and we believe that this evidence further supports the coherency of the goal-generation mechanism: commands assigned to workers (green and red lines) are a function of those sent to sub-managers (blue and orange lines), i.e., the coherency in time among different levels should be preserved. However, goal representations received by left and right sub-managers appear less predictable, and the spectrum of the associated t-SNE embeddings appears much richer and diverse. Thank you for the comment, we will improve the discussion related to this aspect in the revision.
>
> > While this is a well presented and interesting approach to HRL, the paper does not really explore the limitations of HRL in general and this method specifically. There is a sentence referring to the limitations listed in Vezhnevets et al. [2017] but one would imagine that research in HRL would have a different set of concerns more than seven years later.
>
> Our framework is designed to address composable RL tasks and, therefore, limitations must be contextualized within this setting. In this regard, to the best of our knowledge, there is no hierarchical (graph-based) RL model tailored for these problems. Limitations are those of HRL models based on the feudal paradigm, such as FeUdal Networks [4].
> As discussed in Sec. 6 and, more in-depth, in the “Optimization algorithm” paragraph of Sec. 5.1, the main limitation of our approach concerns the current training routine.
>
> > It does not become clear from the manuscript how applicable the proposed method is, e.g., in standard RL environments with challenging temporal credit assignment problems, for which HRL has been proposed as a solution. Discussion of the limitations would help place this work in broader context.
>
> We remark that benchmarks were chosen to highlight the improvements brought by our approach in terms of global coordination and, in this regard, the achieved results support our claims. However, even if not the main focus of the paper, the experiment concerning temporally extended goals of the feudal models (FGNN and FDS) conducted in Sec. 5.2 shows that our approach can mitigate long-term control problems. Future works can address this aspect further and we will make this clear in the paper.
>
>
> **References**:
>
> [1] Wang, Tingwu, et al. "Nervenet: Learning structured policy with graph neural networks." International conference on learning representations. 2018.
>
> [2] Huang, Wenlong, Igor Mordatch, and Deepak Pathak. "One policy to control them all: Shared modular policies for agent-agnostic control." International Conference on Machine Learning. PMLR, 2020.
>
> [3] Kurin, Vitaly, et al. "My Body is a Cage: the Role of Morphology in Graph-Based Incompatible Control." International Conference on Learning Representations. 2020.
>
> [4] Vezhnevets, Alexander Sasha, et al. "Feudal networks for hierarchical reinforcement learning." International conference on machine learning. PMLR, 2017.

---

> > ### Comment · Reviewer_14WH · 2024-11-17
> >
> > Thanks for the detailed response. My main concerns have been addressed.
> >
> > Final remark, consider changing the title of the paper to include "composable RL" or something to the same effect, since the current title doesn't quite narrow down the scope enough.

---

> > > ### Author Response · Authors · 2024-11-18
> > >
> > > Thank you for the feedback!

---

### Review · Reviewer_ELFu · 2024-10-23

**Summary Of Contributions:**

This paper presents a methodology called Feudal Graph Reinforcement Learning. It addresses the limitations of standard graph-based RL methods, which can struggle with global coordination and high-level planning due to information bottlenecks created by local message-passing operators.

FGRL introduces a hierarchical RL approach with a pyramidal message-passing architecture to improve composability in control problems.  The key idea is to establish a hierarchy of policies, mimicking a feudal structure, where high-level commands are propagated down through a layered graph structure. Lower layers in the hierarchy represent the physical system's morphology (like the limbs of a robot), while upper layers represent higher-order sub-modules and overall control.  This hierarchical structure allows for task decomposition, with actions at each level setting goals for the level below.

FGRL was evaluated on a graph clustering problem and MuJoCo locomotion tasks. The experiments showed that the hierarchical message-passing scheme facilitates learning hierarchical decision-making policies and improves coordination, especially in complex tasks requiring higher-level planning.

The main contributions of this paper include:

- They introduce the FGRL paradigm, a new methodological deep learning framework for graph-based HRL in composable environments, and apply it to a graph clustering problem and composable continuous control tasks.
- They provide empirical evidence that supports the adoption of hierarchical message-passing schemes and graph-based representations to implement hierarchical control policies.

**Audience:**

Yes

**Claims And Evidence:**

Yes

**Requested Changes:**

N/A

**Strengths And Weaknesses:**

## Strengths

- The paper is well-written, easy to follow, and the figures are nice.
- Integrating hierarchical structure with graph-based reinforcement learning is a reasonable approach when the task exhibits a specific structure.
- Compared to simpler message-passing methods, the hierarchical message-passing approach improves global coordination and alleviates information bottlenecks common in complex tasks.
- Analysis of the generated goals indicates that the hierarchical structure results in understandable and interpretable command propagation, effectively representing key features of the learned behaviors.

## Weaknesses

- While the proposed method introduces considerable complexity compared to a flat GNN, the empirical evaluation doesn't demonstrate a corresponding significant improvement.  For example, Figure 8 shows the proposed method failing to outperform even a basic MLP agent, which requires neither specialized design nor prior knowledge of the graph structure.  Furthermore, the feasibility of relying on heuristics to create hierarchical graph remains unclear on a borader range of applications.

- The application of the proposed approach to continuous control tasks like MuJoCo is nice.  However, the weak performance is a signifcant issue.  Mainstream reinforcement learning algorithms like SAC achieve substantially better results on Humanoid, Walker, and Half Cheetah environments than the proposed method.  While not every method needs to achieve state-of-the-art performance, results significantly below the community standard raise concerns about the proposed method's practical utility.

---

> ### Author Response · Authors · 2024-10-31
>
> Thank you for the detailed review, please find our point-by-point answers below.
>
> > While the proposed method introduces considerable complexity compared to a flat GNN, the empirical evaluation doesn't demonstrate a corresponding significant improvement. For example, Figure 8 shows the proposed method failing to outperform even a basic MLP agent, which requires neither specialized design nor prior knowledge of the graph structure.
>
> We believe that empirical results do support the adoption of hierarchical graph-based policies. First, in the graph clustering experiments (Sec. 5.2) FGNN drastically outperforms GNN in all but the simplest scenario (refer to Fig. 3). Second, in the MuJoCo benchmarks, FGNN outperforms the baselines in Humanoid and Walker (the agents with the more complex morphology), while obtaining similar performance in Half Cheetah and Hopper. A detailed analysis of these results is reported in Sec. 5.3.  When compared with the MLP baseline, the advantages of FGNN emerge when the complexity of the environment increases, as it performs similarly or better than the baseline in all the MuJoCo benchmarks except for Hopper. Furthermore, the number of parameters for the MLP baseline grows with the number of nodes, which makes it unsuitable for many composable problems such as the graph clustering environment.
>
> > Furthermore, the feasibility of relying on heuristics to create hierarchical graph remains unclear on a borader range of applications.
>
> We remark that when the hierarchical graph cannot be easily extracted using heuristics, graph pooling [1] provides a comprehensive, sound, and well-understood set of operators to build a hierarchy. We will make this clear in the main text.
>
> > The application of the proposed approach to continuous control tasks like MuJoCo is nice. However, the weak performance is a signifcant issue. Mainstream reinforcement learning algorithms like SAC achieve substantially better results on Humanoid, Walker, and Half Cheetah environments than the proposed method. While not every method needs to achieve state-of-the-art performance, results significantly below the community standard raise concerns about the proposed method's practical utility.
>
> We argue that our results are comparable with the performance achieved by PPO in Appendix C.2, showing that the current implementation of our method is already competitive w.r.t. a widely-used gradient-based algorithm. Furthermore, as stated in Sec. 5.3, note that the environments we are using differ from the standard implementation of the MuJoCo benchmarks, meaning that results are not directly comparable to those obtained elsewhere in the literature (see also answer to Reviewer 14WH comments).
> Furthermore, our approach is in principle orthogonal to the policy optimization method (i.e., one might consider combining it with SAC, etc.), but there are technical issues that make this combination not straightforward, as discussed in Sec. 5.1.  Our work represents a first step toward agents based on hierarchies of graph-based policies: future works can build on the framework by extending the approach to more efficient policy optimization methods and consistently achieve state-of-the-art results. We will improve the discussion on these aspects in the conclusion and believe that this is coherent with the scope of the paper.
>
> **References:**
>
> [1] D. Grattarola, D. Zambon, F. M. Bianchi and C. Alippi, "Understanding Pooling in Graph Neural Networks," in IEEE Transactions on Neural Networks and Learning Systems, vol. 35, no. 2, pp. 2708-2718, Feb. 2024, doi: 10.1109/TNNLS.2022.3190922.

---

> ### Comment · Reviewer_ELFu · 2024-11-14
>
> Thanks for the authors' detailed response. However, I still have concerns about the method's applicability if it cannot be integrated with widely-used RL algorithms such as SAC.

---

> > ### Author Response · Authors · 2024-11-15
> >
> > We are sorry if we were not clear enough. Indeed, our framework can be easily integrated with any RL optimization algorithm as each level-specific policy can be trained independently of the others by using any policy optimization method. Nevertheless, as stated in Sec. 5.1, the multi-level structure introduces additional challenges in the learning procedure that off-the-shelf policy optimization methods might not natively handle, but this is another story shared among existing hierarchical architectures. However, we strongly believe this does not compromise the significance and applicability of hierarchical approaches and that future research would solve the issues e.g., possible instabilities coming from the interdependencies among policies learned at each hierarchical level.
> >
> > We recall that our main contribution is the methodological framework for combining graph-based policies with hierarchical RL and is fully validated by the proposed benchmarks and simulations. Therefore, even if the use of gradient-based optimization represents an interesting direction that future works should address, we share the belief that all claims made in the paper are fully supported by evidence.

---

### Review · Reviewer_ktDR · 2024-10-26

**Summary Of Contributions:**

FGRL introduces a layered graph hierarchy, aligning hierarchical reinforcement learning (HRL) principles with graph neural network (GNN) capabilities, and improves task decomposition and coordination for structured environments.

The paper’s primary contributions include:

(1) The FGRL framework, which builds a pyramid-like, hierarchical control structure using GNNs. This structure helps the model break complex tasks into manageable sub-tasks by passing commands through hierarchical levels.

(2) Empirical evaluations on graph clustering and MuJoCo locomotion tasks, showing FGRL’s strong performance against standard baselines.

(3) Analysis supporting hierarchical message-passing as an effective method for learning task-decomposing, decision-making policies.

**Audience:**

Yes

**Claims And Evidence:**

Yes

**Requested Changes:**

Please see the weaknesses.

**Strengths And Weaknesses:**

Strengths:

(1) The integration of feudal (hierarchical) reinforcement learning principles with graph-based modularity introduces an effective structure for decomposing and coordinating tasks across various levels.

(2) The paper demonstrates that hierarchical goals enable long-range coordination and facilitate composable task learning.

Weaknesses:

(1) Constructing a graph-based agent representation may limit the algorithm’s applicability, as it could be restrictive for tasks beyond robotic control.

(2) The baseline comparisons rely on older models (proposed in 2017 and 2018), and the empirical evaluation resembles an ablation study. It omits comparisons with more recent hierarchical RL algorithms, which would provide a more comprehensive evaluation.

(3) The performance results in Figure 5 are notably below state-of-the-art levels, suggesting potential areas for improvement in the algorithm's design. For reference, see the state-of-the-art results reported in https://tianshou.org/en/v0.4.8/tutorials/benchmark.html.

---

> ### Author Response · Authors · 2024-10-31
>
> Thank you for the detailed review, please find our point-by-point answers below.
>
> > Constructing a graph-based agent representation may limit the algorithm’s applicability, as it could be restrictive for tasks beyond robotic control.
>
> Our framework is designed for tackling composable problems and we argue that, in this context, graphs often emerge as a natural representation of such agents. Indeed, there are many possible applications other than robotic tasks for which our framework can be exploited, e.g., power grid management [1,2] or traffic signal control [3].
>
> > The baseline comparisons rely on older models (proposed in 2017 and 2018), and the empirical evaluation resembles an ablation study. It omits comparisons with more recent hierarchical RL algorithms, which would provide a more comprehensive evaluation.
>
> Our proposed approach is tailored to composable control and aims to address the limitations of flat GNNs (e.g., NerveNet [4]). For this reason, the comparison with such architecture is crucial for supporting our claims. Concerning hierarchical algorithms, we remark that our FDS architecture can be regarded as an adaptation of a hierarchical model (e.g., FuN [5]) to the composable setting, and, to the best of our knowledge, there is no HRL method designed for tackling such problems. Therefore, we believe that these considerations make the proposed benchmarks and baselines appropriate.
>
> > The performance results in Figure 5 are notably below state-of-the-art levels, suggesting potential areas for improvement in the algorithm's design. For reference, see the state-of-the-art results reported in https://tianshou.org/en/v0.4.8/tutorials/benchmark.html.
>
> We agree with the reviewer that there is room for improvements in future works, specifically in the optimization algorithm, as we also pointed out in Sec. 5.1 and Sec. 6. However, as stated in the main text (Sec. 5.3), it is important to notice that the environments we use differ from the standard MuJoCo benchmarks used elsewhere in the RL literature, making results not directly comparable (see also answer to Reviewer 14WH). In Appendix C.2 we provide a comparison with a standard gradient-based algorithm such as PPO in the composable setting: our method achieves competitive returns w.r.t. PPO and, at the same time, allows the design of policies with desirable properties for composable problems, such as modularity, transferability, and scalability.
>
> **References:**
>
> [1] Yoon, Deunsol, et al. "Winning the l2rpn challenge: Power grid management via semi-markov afterstate actor-critic." International Conference on Learning Representations. 2021.
>
> [2] Hassouna, Mohamed, et al. "Graph Reinforcement Learning in Power Grids: A Survey." arXiv preprint arXiv:2407.04522 (2024).
>
> [3] Yoon, Jinwon, et al. "Transferable traffic signal control: Reinforcement learning with graph centric state representation." Transportation Research Part C: Emerging Technologies 130 (2021): 103321.
>
> [4] Wang, Tingwu, et al. "Nervenet: Learning structured policy with graph neural networks." International conference on learning representations. 2018.
>
> [5] Vezhnevets, Alexander Sasha, et al. "Feudal networks for hierarchical reinforcement learning." International conference on machine learning. PMLR, 2017.

---

### Author Response · Authors · 2024-11-05

We thank the reviewers again for their feedback. We have attached an improved version of the paper according to the requested changes (edits are reported in blue).

Since we noticed that one of the main concerns is related to the comparison with state-of-the-art methods, we further discuss this aspect here in a general comment, summarizing what was discussed in the individual answers to each review. Results achieved in the MuJoCo benchmarks cannot be directly compared with those reported by many papers in the literature. Indeed, we used a modular variant of these environments taken from Huang et al [1], and, besides changes to the observation space, the version of the physical simulator might also be different depending on the MuJoco version of the environment. Nevertheless, we provide a comparison with a widely-used policy gradient algorithm in Appendix C.2. We also comment that even if the hierarchical setup introduces additional challenges in the training procedure (see Sec. 5.1 and Sec. 6), our method is, in principle, orthogonal to the underlying optimization routine.  However, the combination with existing policy gradient methods is not straightforward and out of the scope of the present paper.  In this aspect, our work provides a foundation for future works to build upon.


**References:**

[1] Huang, Wenlong, Igor Mordatch, and Deepak Pathak. "One policy to control them all: Shared modular policies for agent-agnostic control." International Conference on Machine Learning. PMLR, 2020.

---

### Author Response · Authors · 2024-11-07

We thank again the reviewers for their valuable feedback. Since the author-reviewer discussion period is almost over, we wanted to check whether the questions raised in the reviews have been addressed. If not, we are willing to provide further comments and clarifications.

---

### Decision · Action_Editor_ZovL · 2024-11-26

**Recommendation:** Accept as is

**Comment:**

Reviewers are mostly satisfied with responses and adjustments made in response to their reviews, and in favour of acceptance.

There is one remaining concern revolving around choice of baselines to compare to (e.g., no SAC) and the strength of empirical results. While stronger results and more baselines would of course never be bad, claims and conclusions are already carefully phrased, and the experiments as they are now adequately support them. Therefore, I do not view this as a reason for rejection.

For the camera-ready version, please try to include source code (which, at this point, need not be anonymous anymore), as promised in Appendix D.1.

**Audience:**

Yes.

**Claims And Evidence:**

Yes.